# The Light Dependence of Mesophyll Conductance and Relative Limitations on Photosynthesis in Evergreen Sclerophyllous *Rhododendron* Species

**DOI:** 10.3390/plants9111536

**Published:** 2020-11-10

**Authors:** Ying-Jie Yang, Hong Hu, Wei Huang

**Affiliations:** 1Kunming Institute of Botany, Chinese Academy of Sciences, Kunming 650201, China; yangyingjie@mail.kib.ac.cn; 2University of Chinese Academy of Sciences, Beijing 100049, China

**Keywords:** CO_2_ diffusion, chloroplast CO_2_ concentration, photorespiration, photosynthesis, relative limitations, *Rhododendron*

## Abstract

Mesophyll conductance (*g*_m_) limits CO_2_ diffusion from sub-stomatal internal cavities to the sites of RuBP carboxylation. However, the response of *g*_m_ to light intensity remains controversial. Furthermore, little is known about the light response of relative mesophyll conductance limitation (*l*_m_) and its effect on photosynthesis. In this study, we measured chlorophyll fluorescence and gas exchange in nine evergreen sclerophyllous *Rhododendron* species. *g*_m_ was maintained stable across light intensities from 300 to 1500 μmol photons m^−2^ s^−1^ in all these species, indicating that *g*_m_ did not respond to the change in illumination in them. With an increase in light intensity, *l*_m_ gradually increased, making *g*_m_ the major limiting factor for area-based photosynthesis (*A*_N_) under saturating light. A strong negative relationship between *l*_m_ and *A*_N_ was found at 300 μmol photons m^−2^ s^−1^ but disappeared at 1500 μmol photons m^−2^ s^−1^, suggesting an important role for *l*_m_ in determining *A*_N_ at sub-saturating light. Furthermore, the light-dependent increase in *l*_m_ led to a decrease in chloroplast CO_2_ concentration (*C*_c_), inducing the gradual increase of photorespiration. A higher *l*_m_ under saturating light made *A*_N_ more limited by RuBP carboxylation. These results indicate that the light response of *l*_m_ plays significant roles in determining *C*_c_, photorespiration, and the rate-limiting step of *A*_N_.

## 1. Introduction

In southwest China, mountain forest ecosystems are characterized by the presence of evergreen sclerophyllous angiosperms. This vegetation type includes many species of the genus *Rhododendron* (Ericaceae) that exhibit low mesophyll conductance (*g*_m_) [1]. Leaf CO_2_ diffusion mainly includes diffusion from air to the intercellular cavity (stomatal conductance, *g*_s_) and diffusion from the intercellular cavity to the site of RuBP carboxylation (*g*_m_). Many previous studies indicated that sclerophyllous plants have relatively lower *g*_m_ values than herbaceous plants [1,2,3,4,5]. Such low levels of *g*_m_ in sclerophyllous plants limit *A*_N_ to a large extent when exposed to high light. Consequently, *g*_m_ is the major limiting factor for *A*_N_ under constant high light in sclerophyllous tree species, such as Mediterranean evergreens [4,6,7,8], the sclerophyllous genus *Banksia* [2], *Eucalyptus camaldulensis* [9,10], and sclerophyllous *Rhododendron* species [1]. Therefore, the response of *g*_m_ to the environment plays a critical role in controlling photosynthesis in sclerophyllous angiosperms.

Under natural field conditions, leaves experience rapid changes in light intensities on timescales of seconds or minutes [11,12]. *g*_s_ usually increases with increasing light in angiosperms, which raises the question whether *g*_m_ changes rapidly in response to the change in light intensity. Some studies observed that *g*_m_ increased with increasing irradiance in *Eucalyptus* species [13,14], *Nicotiana tabacum* [15], *Camellia* species [16], *Triticum durum* and *Arbutus* × ‘Marina’ [17], and rice (*Oryza sativa*) grown under high nitrogen concentrations [18]. Furthermore, a recent study found that with increasing irradiance, *g*_m_ gradually increased in all six studied angiosperms [19]. In contrast, other studies reported that light intensity did not affect *g*_m_ in *Triticum aestivum* [20], *N. tabacum* [21], and rice grown under low nitrogen concentrations [18]. Therefore, the response of *g*_m_ to increases in light intensity in angiosperms is species dependent and can be affected by nitrogen nutritional conditions. Previous studies mainly focused on the light response of *g*_m_ in herbaceous plants. However, the response of *g*_m_ to light intensity in sclerophyllous evergreen species is not well known.

Many previous studies have analyzed the quantitative relative stomatal, mesophyll, and biochemical limitations of *A*_N_ under saturating light conditions [4,22,23,24,25]. As we know, the value of *A*_N_ under saturating light largely determines the growth rate of plants [26,27,28]. However, some shade-tolerant plant species and leaves in lower parts of canopies may experience moderate light. As a result, the value of *A*_N_ at moderate light can significantly affect plant growth and crop productivity [29]. Therefore, understanding the relative limitations of *A*_N_ at moderate light may have broad applications in angiosperms and crops in particular. However, light response changes in the relative limitations of *A*_N_ are poorly understood.

In addition to RuBP carboxylation, Rubisco catalyzes RuBP oxygenation under current atmospheric environmental conditions [30,31,32]. The photorespiratory pathway converts phosphoglycolate (2PG) to 3-phosphoglycerate (3PGA), allowing the Calvin–Benson cycle to operate in the presence of molecular oxygen [33,34]. Under low light, *A*_N_ is mainly limited by a lack of light energy, and the resulting high chloroplast CO_2_ concentration (*C*_c_) restricts the rate of RuBP oxygenation (*V*_o_). Under high light, the increased rate of RuBP carboxylation (*V*_c_) leads to a decrease in *C*_c_ [9,21], increasing the *V*_o_/*V*_c_ ratio and thus enhancing photorespiration [35,36]. Therefore, photorespiration usually increases with increasing light intensity in C_3_ plants [16,37,38,39,40]. Meanwhile, either an increase in *g*_s_ or *g*_m_ can partially compensate for the CO_2_ consumption in CO_2_ fixation. However, it is unclear whether the light response of photorespiration is mainly determined by *g*_s_ or *g*_m_.

At saturating light, *A*_N_ can be limited by RuBP carboxylation and/or RuBP regeneration [41,42,43]. Once the operating *C*_c_ is lower than the chloroplast CO_2_ concentration (*C*_trans_) at which the transition from RuBP carboxylation limitation to RuBP regeneration limitation occurs, *A*_N_ is limited by RuBP carboxylation. When *C*_c_ is higher than *C*_trans_, then *A*_N_ tends to be limited by RuBP regeneration. The major rate-limiting step of *A*_N_ is species dependent and can be affected by leaf nitrogen content and measurement temperature [21,41,43]. However, the effect of *g*_m_ on the rate-limiting step of *A*_N_ is poorly understood. The leaf *g*_m_ is positively correlated to leaf nitrogen content [18,38,43]. Furthermore, *A*_N_ tends to be limited by RuBP regeneration in plants grown under high nitrogen concentrations but is limited by RuBP carboxylation in plants grown under nitrogen-deficient concentrations [43]. Therefore, we hypothesize that the rate-limiting step of *A*_N_ under saturating light is largely determined by *g*_m_.

In this study, we measured light responses of gas exchange and chlorophyll fluorescence in nine evergreen sclerophyllous *Rhododendron* species. The aims of this study were (1) to investigate the light response changes in *g*_m_ and the relative limitations of *A*_N_; (2) to assess whether the increase in *V*_o_/*V*_c_ ratio under high light is mainly determined by *g*_s_ or *g*_m_; and (3) to examine the effects of *g*_m_ and *l*_m_ on the rate-limiting step of *A*_N_.

## 2. Results

### 2.1. Light Intensity Dependence of Photosynthesis and Mesophyll Conductance

The *A*_800_/*A*_1500_ ratios ranged from 0.86 to 1.02, suggesting that *A*_N_ were saturated or almost saturated at 800 μmol photons m^−2^ s^−1^. Under a high light of 1500 μmol photons m^−2^ s^−1^, *A*_N_ were saturated in all *Rhododendron* species. The light-saturated *A*_N_ ranged from 10.3 (*R. ciliicalyx*) to 17.8 μmol CO_2_ m^−2^ s^−1^ (*R. glanduliferum*), a total variation of 60% between species (Figure 1). Under such saturating light, the PSII electron transport rate (*J*_PSII_) ranged from 151.9 (*R. decorum* subsp. *diaprepes*) to 205.2 μmol electrons m^−2^ s^−1^ (*R. delavayi*), leading to a variation of 35% in *J*_PSII_ (Figure 1).

We calculated *g*_m_ under different light intensities according to the method of [44]. The value of *g*_m_ at 300 μmol photons m^−2^ s^−1^ varied between species, ranging from 0.040 (*R. ciliicalyx*) to 0.20 mol m^−2^ s^−1^ (*R. decorum* subsp. *diaprepes*) (Figure 2). At 1500 μmol photons m^−2^ s^−1^, *g*_m_ ranged from 0.049 (*R. ciliicalyx*) to 0.16 mol m^−2^ s^−1^ (*R. decorum* subsp. *diaprepes*) (Figure 2). All species showed no significant change in *g*_m_ between 300 and 1500 μmol photons m^−2^ s^−1^ and *g*_m_ was maintained stable over the light intensity change (Figure 2). The average light response values of *g*_m_ ranged from 0.048 (*R. ciliicalyx*) to 0.18 mol m^−2^ s^−1^ (*R. decorum* subsp. *diaprepes*) (Figure 2). 

### 2.2. Light Intensity Dependence of Relative Limitations of Photosynthesis

The relative limitations of photosynthesis were significantly affected by the light intensity. Under low light, the rate of photosynthesis was largely limited by biochemical capacity (*l*_b_) (Figure 3), owing to the lack of ATP and NADPH. With an increase in light intensity, *l*_b_ gradually decreased (Figure 3). Meanwhile, mesophyll conductance limitation (*l*_m_) gradually increased and stomatal conductance limitation (*l*_s_) changed slightly (Figure 3). At a moderate light intensity of 500 μmol photons m^−2^ s^−1^, the major limiting factor for photosynthesis shifted from *l*_b_ to *l*_m_, except for the species with the highest *g*_m_: *R. decorum* subsp. *diaprepes* (Figure 3). Under the saturating light of 1500 μmol photons m^−2^ s^−1^, *l*_m_ ranged from 0.37 to 0.71, *l*_b_ from 0.15 to 0.37, and *l*_s_ from 0.12 to 0.26 (Figure 3). Therefore, *g*_m_ is the major limiting factor for CO_2_ assimilation at saturating light in *Rhododendron* species, followed by biochemical capacity and stomatal conductance. 

The relationships between *g*_m_, *l*_m_, and *A*_N_ at sub-saturating and saturating light intensities were also analyzed in these studied species (Figure 4). As expected, the values of *g*_m_ were significantly positively correlated to *A*_N_. Interestingly, a closer relationship between *g*_m_ and *A*_N_ was found at 300 μmol photons m^−2^ s^−1^ than that at 1500 μmol photons m^−2^ s^−1^ (Figure 4A). Furthermore, a close negative correlation was found between *l*_m_ and *A*_N_ at 300 μmol photons m^−2^ s^−1^ (Figure 4B). However, this significant relationship disappeared at 1500 μmol photons m^−2^ s^−1^ (Figure 4B). These results indicate that *g*_m_ and *l*_m_ play more important roles in determining *A*_N_ at sub-saturating light than at saturating light. 

### 2.3. Light Intensity Dependence of Chloroplast CO_2_ Concentration and Photorespiration

With an increase in light intensity, intercellular CO_2_ concentration (*C*_i_) and chloroplast CO_2_ concentration (*C*_c_) gradually decreased in all *Rhododendron* species (Figure 5A,B). At 1500 μmol photons m^−2^ s^−1^, *C*_i_ ranged from 253 (*R. decorum* subsp. *diaprepes*) to 317 μmol mol^−1^ (*R. glanduliferum*) (Figure 5A), and *C*_c_ ranged from 76 (*R. hancockii*) to 144 μmol mol^−1^ (*R. decorum* subsp. *diaprepes*) (Figure 5B). Further analysis found that the drop in *C*_c_ was tightly correlated with an increase in *l*_m_ (Figure 5C). Therefore, the light dependence change in *C*_c_ was mainly caused by an increase in *l*_m_. *C*_c_ is known to be a key factor affecting the affinity of Rubisco to CO_2_ and O_2_, and thus determines the value of *V*_o_/*V*_c_. With an increase in illumination, the *V*_o_/*V*_c_ ratio gradually increased (Figure 6A). Furthermore, the light dependence of *V*_o_/*V*_c_ was positively correlated to *l*_m_ (Figure 6B). These results indicate that the increased *V*_o_/*V*_c_ with increasing irradiance was mainly caused by the enhanced *l*_m_. 

At 1500 μmol photons m^−2^ s^−1^, *C*_trans_ was calculated to analyze the rate-limiting step of *A*_N_. The operating *C*_c_ values are significantly lower than the *C*_trans_ values in these studied species, except for *R. decorum* subsp. *diaprepes* (Figure 7). Therefore, the saturating *A*_N_ at current atmospheric CO_2_ concentration was mainly limited by RuBP carboxylation in these *Rhododendron* species. In *R. decorum* subsp. *diaprepes*, *A*_N_ was limited by RuBP carboxylation and regeneration. 

## 3. Discussion

### 3.1. Rapid Response of g_m_ to Changes in Light Intensity is Consistent among Rhododendron Species

With an increase in light intensity, stomatal conductance gradually increased to enhance the CO_2_ diffusion from air to sub-stomatal internal cavities and thus to favor photosynthetic CO_2_ assimilation [45,46,47]. Subsequently, mesophyll conductance limits the diffusion of CO_2_ from intercellular cavities to the sites of carboxylation, and may respond rapidly to changes in light intensity. Some previous studies observed that *g*_m_ rapidly increased with increasing irradiance in *N. tabacum* [15], *Triticum durum*, and *Arbutus* × ‘Marina’ [17], and three *Eucalyptus* species *E. globules*, *E. saligna*, and *E. sieberi* [13,14]. A recent study reported that *g*_m_ rapidly responded to changes in light intensity in six studied angiosperms [19]. By contrast, some authors found that *g*_m_ was not responsive to irradiance in *Triticum aestivum* [20] and *N. tabacum* [21]. Therefore, the response of *g*_m_ to light intensity highly differs between species, presenting an important photosynthetic response to environmental change. However, the light response of *g*_m_ in evergreen sclerophyllous angiosperms is poorly understood.

This article investigated the rapid response of *g*_m_ to light intensity in nine sclerophyllous *Rhododendron* species. We found that the values of *g*_m_ did not change between 300 and 1500 μmol photons m^−2^ s^−1^ in either of these species (Figure 2). Therefore, these nine *Rhododendron* species showed the same response model of *g*_m_ to changes in light intensity. Consistently, the three *Eucalyptus* species *E. globules*, *E. saligna*, and *E. sieberi* also showed the same trend in the light response of *g*_m_ [13,14]. Therefore, we propose that the rapid response of *g*_m_ to the change in light intensity may be a conservative photosynthetic trait in a given genus. This conclusion is applicable only under normal growth conditions because water and/or nutrition stresses can alter the rapid response of *g*_m_ to irradiance.

### 3.2. Light Response Changes in Relative Limitations of Photosynthesis

Many studies have documented that plant growth and crop productivity are largely linked to *A*_N_ under high light [26,27,28]. However, global rice productivity in agricultural fields was not determined by the maximum *A*_N_ under saturating light but was linked to the *A*_N_ under low light [29]. Therefore, in order to optimize the cultivation strategy, we should take into consideration the relative limitations of photosynthesis under different light intensities. While the relative limitations of *A*_N_ under high light are widely studied, information on the light response of relative limitations is very limited. In this article, we analyzed the relative limitations of *A*_N_ under different light intensities in nine *Rhododendron* species. The stomatal conductance limitation was the most insignificant factor for *A*_N_ in these species, irrespective of light intensity (Figure 3). Under low light, the light-dependent production of ATP and NADPH was restricted by limiting light energy, and *A*_N_ was mainly limited by biochemical factors (Figure 3). With an increase in light intensity, ETRII rapidly increased (Figure 1), leading to the increased production rates of ATP and NADPH. In contrast, the value of *C*_c_ gradually decreased (Figure 5B). Therefore, the change in *C*_c_ was out-of-step with the change in energy supply, generating an imbalance between CO_2_ supply and production of ATP and NADPH. When exposed to a moderate light of 500 μmol photons m^−2^ s^−1^, the most limiting factor for *A*_N_ shifted from biochemical capacity to *g*_m_ (Figure 3). Therefore, *g*_m_ was the major limiting factor for *A*_N_ in these species when exposed to high light. Furthermore, the values of *A*_N_ at 300 μmol photons m^−2^ s^−1^ were largely correlated to the values of *g*_m_ and *l*_m_ (Figure 4), indicating that increasing *g*_m_ has a significant potential to increase *A*_N_ under sub-saturating light. Taking into consideration that different plants have different light saturating points, measuring the light dependence changes in relative limitations of *A*_N_ may have broad applications in tree breeding and crop improvement. 

With an increase in light intensity, electron transport for photorespiration gradually increased [16,37,38,39]. However, it is unclear whether the light response of photorespiration was caused by the change of *l*_m_ or *l*_s_. In this study, we found that *l*_s_ showed only small responses to changes in light intensity (Figure 3). By comparison, *l*_m_ gradually increased with light intensity (Figure 3). Meanwhile, *C*_c_ gradually decreased (Figure 5B) and the ratio of *V*_o_/*V*_c_ gradually increased (Figure 6A). Furthermore, with an increase in light intensity, the decrease in *C*_c_ was tightly correlated to an increased *l*_m_ (Figure 5C). Thus, the stable *g*_m_ was insufficient to compensate for CO_2_ consumption by *A*_N_ under high light, increasing the affinity of Rubisco to O_2_, and thus enhancing photorespiration. Therefore, the light response of photorespiration is more determined by *l*_m_ rather than *l*_s_ (Figure 6B).

### 3.3. CO_2_ Assimilation under High Light Tends to be Limited by RuBP Carboxylation in Rhododendron Species

In C_3_ plants, *A*_N_ under saturating light can be limited by RuBP carboxylation and/or RuBP regeneration [21,42,43]. Once the value of *C*_c_ is lower than the value of *C*_trans_, *A*_N_ tends to be limited by RuBP carboxylation [43]. When *C*_c_ is higher than *C*_trans_, *A*_N_ is then limited by RuBP regeneration. However, the rate-limiting step of *A*_N_ in sclerophyllous species is poorly understood. In this study, we found that the values of operating *C*_c_ under saturating light were significantly lower than the values of *C*_trans_ in these species, apart from in *R. decorum* subsp. *disprepes* (Figure 7). Therefore, *A*_N_ under saturating light in these *Rhododendron* species was mainly limited by RuBP carboxylation. 

In general, the rate-limiting step of *A*_N_ can be influenced by temperature and leaf nitrogen content [41,43]. For example, at normal growth temperatures, *A*_N_ is limited by RuBP carboxylation under nitrogen deficiency but tends to be limited by RuBP regeneration at high nitrogen conditions in C_3_ crop species [43]. In this article, we found that the rate-limiting step of *A*_N_ by RuBP carboxylation in *Rhododendron* species was mainly caused by a relatively lower *g*_m_ (Figure 2 and Figure 7). In rice (*Oryza sativa*), wheat (*Triticum aestivum*), spinach (*Spinacia oleracea*), and tobacco (*Nicotiana tabacum*), values for *g*_m_ at saturating light were higher than 0.2 mol m^−2^ s^−1^ [43,48]. By comparison, the studied *Rhododendron* species displayed *g*_m_ values ranging from 0.048 to 0.18 mol m^−2^ s^−1^ (Figure 2). Such low *g*_m_ in these *Rhododendron* species limited the diffusion of CO_2_ from intercellular cavities to chloroplasts, resulting in *C*_c_ being lower than *C*_trans_. Therefore, *g*_m_ has the potential to alter the rate-limiting step of *A*_N_ under high light.

## 4. Materials and Methods

### 4.1. Plant Materials and Growth Conditions

Nine evergreen sclerophyllous *Rhododendron* species from China were studied: *R*. *decorum*, *R*. *glanduliferum*, *R*. *hancockii*, *R. decorum* subsp. *disprepes*, *R*. *ciliicalyx*, *R. delavayi*, *R. davidii*, *R. fortunei*, and *R. pachypodum*. All plants of these species are cultivated in a common garden at the Kunming Botanical Garden, Yunnan, China (102°44′31″ E, 25°08′24″ N, 1950 m of elevation). We chose fully expanded but not senescent sun leaves for photosynthetic measurements. For each species, at least four leaves from different individual plants were measured. 

### 4.2. Gas Exchange and Chlorophyll Fluorescence Measurements

Gas exchange and chlorophyll fluorescence were recorded using the 2-cm^2^ measuring head of LI-6400XT (Li-Cor Biosciences, Lincoln, NE, USA). All measurements were conducted at approximately 25 °C and relative air humidity near 60%. After photosynthetic induction at 1500 μmol photons m^−2^ s^−1^ for 20 min, light response curves were recorded at a 400 μmol mol^−1^ CO_2_ concentration, and photosynthetic parameters were monitored after exposure to each light intensity for 2 min. After light adaptation at 1500 μmol photons m^−2^ s^−1^ and 400 μmol mol^−1^ CO_2_ concentration for 20 min, *A*/*C*_i_ measurements were made at 50, 100, 150, 200, 300, 400, 600, 800, 1000, and 1200 μmol mol^−1^ CO_2_ concentrations. For each CO_2_ concentration, photosynthetic measurement was completed in 2 to 3 min. Using the *A*/*C*_i_ curves, the maximum rates of RuBP regeneration (*J*_max_) and carboxylation (*V*_cmax_) were calculated [49].

The quantum yield of photosystem II (PSII) photochemistry was calculated as Φ_PSII_ = (*F_m_’* − *F_s_*)/*F_m_’* [50], where *F_m_’* and *F_s_* represent the maximum and steady-state fluorescence after light adaption, respectively [51]. The total electron transport rate through PSII (*J*_PSII_) was calculated as follows [52]:$$J_{\mathrm{PSII}}=\Phi_{\mathrm{PSII}}\times \mathrm{PPFD}\times L_{\mathrm{abs}}\times 0.5,$$
where PPFD is the photosynthetic photon flux density and leaf absorbance (*L*_abs_) is assumed to be 0.84. We applied the constant of 0.5 based on the assumption that photons were equally distributed between PSI and PSII. 

### 4.3. Estimation of Mesophyll Conductance and Chloroplast CO_2_ Concentration

We calculated mesophyll conductance according to the following equation [44]:$$g_{m}=\frac{A_{N}}{C_{i}-\Gamma^{*}\left(J_{\mathrm{PSII}}+8\left(A_{N}+R_{d}\right)\right)/\left(J_{\mathrm{PSII}}-4\left(A_{N}+R_{d}\right)\right)},$$
where *A*_N_ represents the net rate of CO_2_ assimilation; *C*_i_ is the intercellular CO_2_ concentration; and Γ* is the CO_2_ compensation point in the absence of daytime respiration [41,48], and we used a typical value of 40 umol mol^−1^ in our current study [19]. Respiration rate in the dark (*R*_d_) was considered to be the half of the dark-adapted mitochondrial respiration rate as measured after 10 min of dark adaptation [23].

Based on the estimated *g*_m_, we then calculated the chloroplast CO_2_ concentration (*C*_c_) according to the following equation [49,53]: $$C_{c}=C_{i}-\frac{A_{N}}{g_{m}}.$$

To identify the rate-limiting step of CO_2_ assimilation, we subsequently estimated *C*_trans_ (the chloroplast CO_2_ concentration at which the transition from RuBP carboxylation to RuBP regeneration occurred) [21,43]:$$C_{\mathrm{trans}}=\;\frac{K_{c}\left(1+O/K_{o}\right)J_{\max}/4V_{\mathrm{cmax}}-2\Gamma^{*}}{1-J_{\max}/4V_{\mathrm{cmax}}}$$
where *K*_c_ (μmol mol^−1^) and *K*_o_ (mmol mol^−1^) are assumed to be 407 μmol mol^−1^ and 277 mmol mol^−1^ at 25 °C, respectively (Long and Bernacchi 2003); *O* was assumed to be 210 mmol mol^−1^ (Farquhar et al., 1980). The rate-limiting step for CO_2_ assimilation was analyzed by comparing the values of *C*_c_ and *C*_trans_. *A*_N_ tends to be limited by RuBP carboxylation when *C*_c_ is lower than *C*_trans_ and tends to be limited by RuBP regeneration when *C*_c_ is higher than *C*_trans_.

### 4.4. Quantitative Limitation Analysis of A_N_

Relative photosynthetic limitations were assessed as follows [22]:$$l_{s}=\frac{g_{\mathrm{tot}}/g_{s}\times \partial A_{N}/\partial C_{c}}{g_{\mathrm{tot}}+\partial A_{N}/\partial C_{c}}$$
$$l_{m}=\frac{g_{\mathrm{tot}}/g_{m}\times \partial A_{N}/\partial C_{c}}{g_{\mathrm{tot}}+\partial A_{N}/\partial C_{c}}$$
$$l_{b}=\frac{g_{\mathrm{tot}}}{g_{\mathrm{tot}}+\partial A_{N}/\partial C_{c}}$$
where *l*_s_, *l*_m_, and *l*_b_ represent the relative limitations of stomatal conductance, mesophyll conductance, and biochemical capacity, respectively, in setting the observed value of *A*_N_. *g*_tot_ is the total conductance of CO_2_ between the leaf surface and sites of RuBP carboxylation (calculated as 1/*g*_tot_ = 1/*g*_s_ + 1/*g*_m_). 

### 4.5. Modeling of V_c_ and V_o_

The rates of RuBP carboxylation (*V*_c_) and oxygenation (*V*_o_) were calculated as follows [36]: $$V_{c}=\frac{A_{n}+R_{d}}{1-\left(\Gamma^{*}/C_{c}\right)}$$
and:$$V_{o}=\frac{A_{n}+R_{d}}{\left(C_{c}/2\Gamma^{*}\right)-0.5}$$

### 4.6. Statistical Analysis

Data were displayed as means ± SE (*n* = 4–5). After testing for normality and homogeneity of variances, one-Way ANOVA tests were used at *α* = 0.05 significance level to determine whether significant differences existed between different averages.

## 5. Conclusions

In this study, we examined the light response of *g*_m_ and its effect on photosynthesis in nine evergreen sclerophyllous *Rhododendron* species. The results indicated that all species showed no significant response of *g*_m_ to variations in illumination. Therefore, the response of *g*_m_ to rapid changes in irradiance may be a conservative photosynthetic trait in a given genus, at least in the genus *Rhododendron*. Furthermore, we found that the light response of photorespiration was mainly determined by *g*_m_ limitation rather than *g*_s_ limitation. At saturating light, *g*_m_ limitation significantly affected the differentials between *C*_trans_ and *C*_c_, thus altering the rate-limiting step of *A*_N_. We propose that examining the light dependence changes in relative limitations of *A*_N_ can provide some valuable means for tree breeding and crop improvement.

## Figures and Tables

**Figure 1 plants-09-01536-f001:**
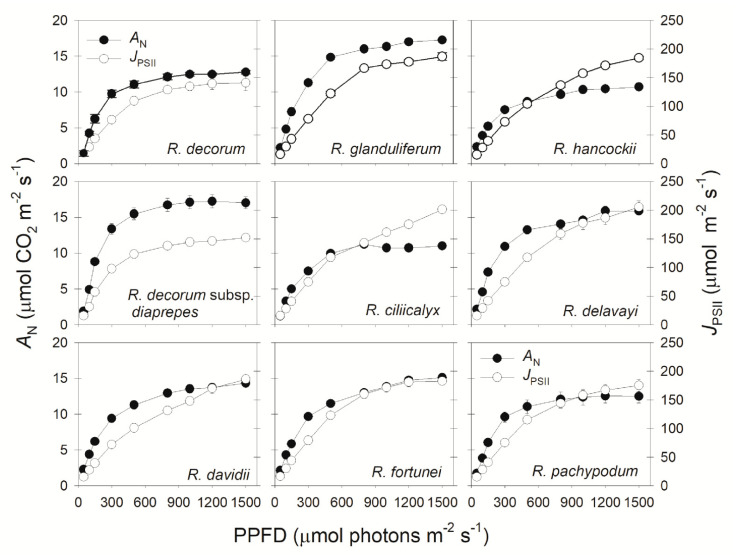
Responses of the leaf CO_2_ assimilation rate (*A*_N_) and electron transport rate (*J*_PSII_) to incident photosynthetic photon flux density (PPFD) in nine *Rhododendron* species. Symbols represent means ± SE (*n* = 4–5).

**Figure 2 plants-09-01536-f002:**
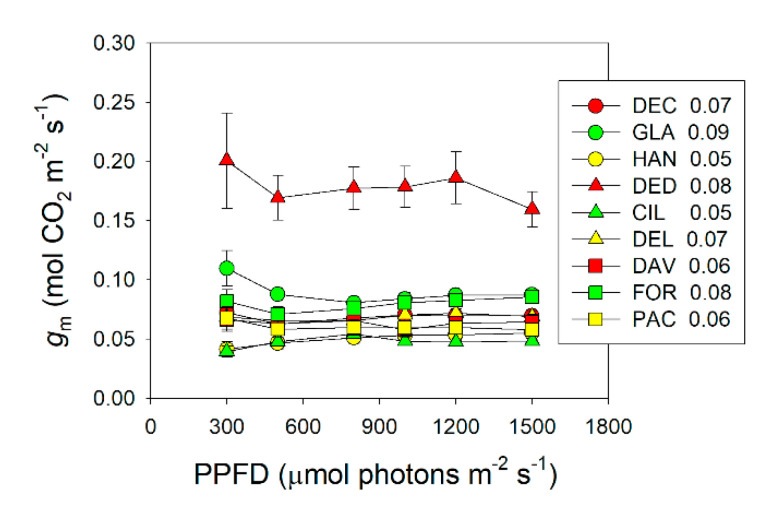
Response of mesophyll conductance (*g*_m_) to incident light intensity in nine *Rhododendron* species. The average *g*_m_ values across light response curves in these nine *Rhododendron* species are displayed with the symbols. Symbols represent means ± SE (*n* = 4–5). Species abbreviations: DEC, *R. decorum*; GLA, *R. glanduliferum*; HAN, *R. hancockii*; DED, *R. decorum* subsp. *disprepes*; CIL, *R. ciliicalyx*; DEL, *R. delavayi*; DAV, *R. davidii*; FOR, *R. fortunei*; PAC, *R. pachypodum*.

**Figure 3 plants-09-01536-f003:**
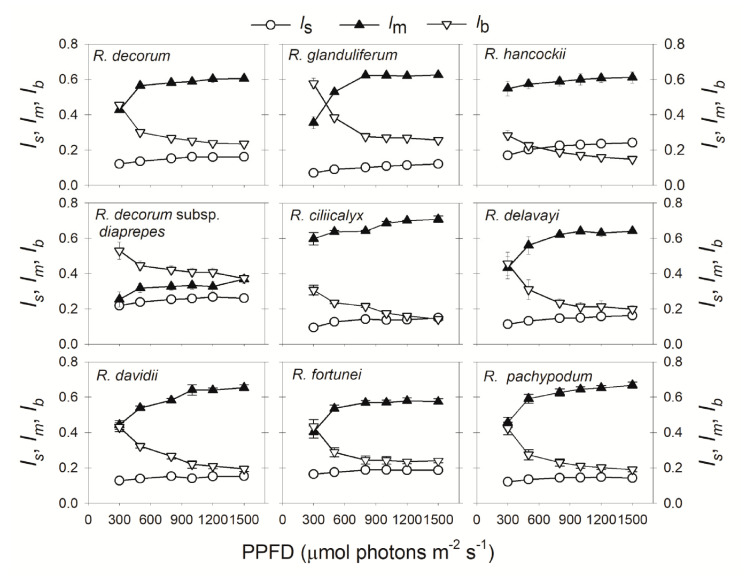
Light intensity dependence of relative limitations of photosynthesis in nine *Rhododendron* species. *l*_s_, relative stomatal conductance limitation; *l*_m_, relative mesophyll conductance limitation; *l*_b_, relative biochemical limitation. Symbols represent means ± SE (*n* = 4–5).

**Figure 4 plants-09-01536-f004:**
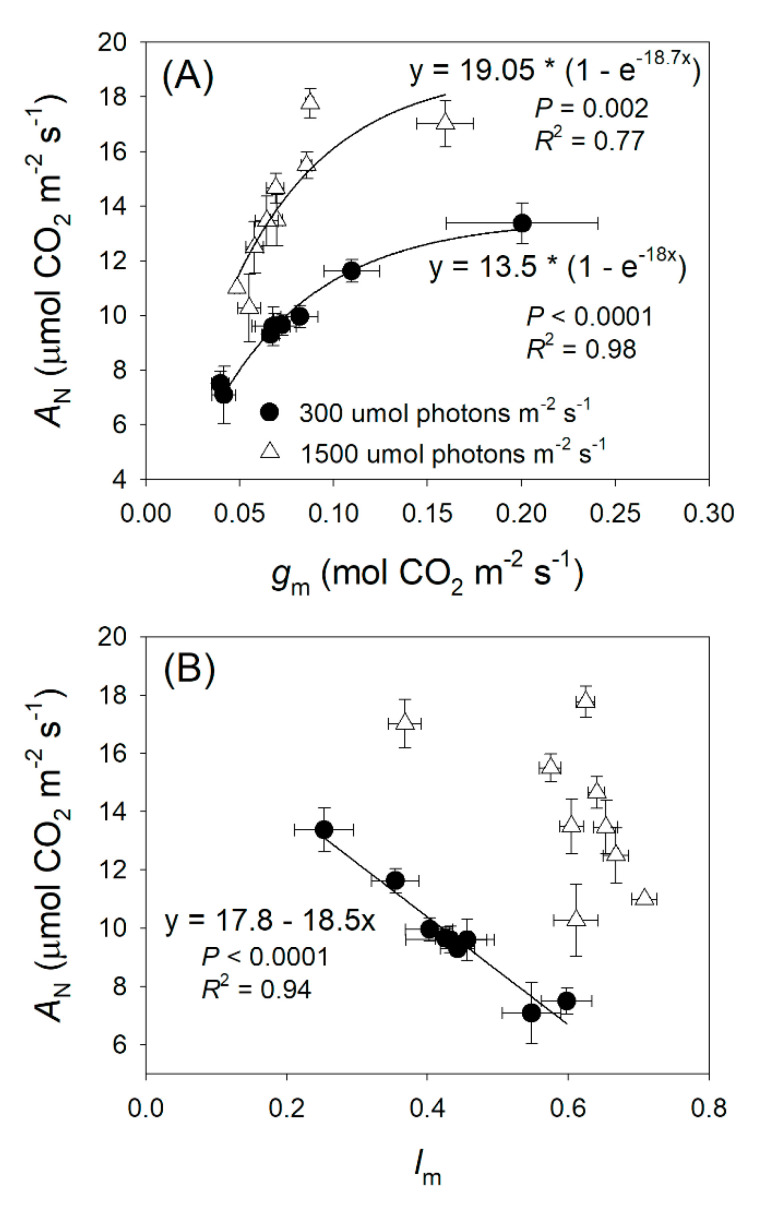
The relationship between *g*_m_ and *A*_N_ (**A**), *l*_m_ and *A*_N_ (**B**) at 300 and 1500 μmol photons m^−2^ s^−1^ in nine *Rhododendron* species. Symbols represent means ± SE (*n* = 4–5).

**Figure 5 plants-09-01536-f005:**
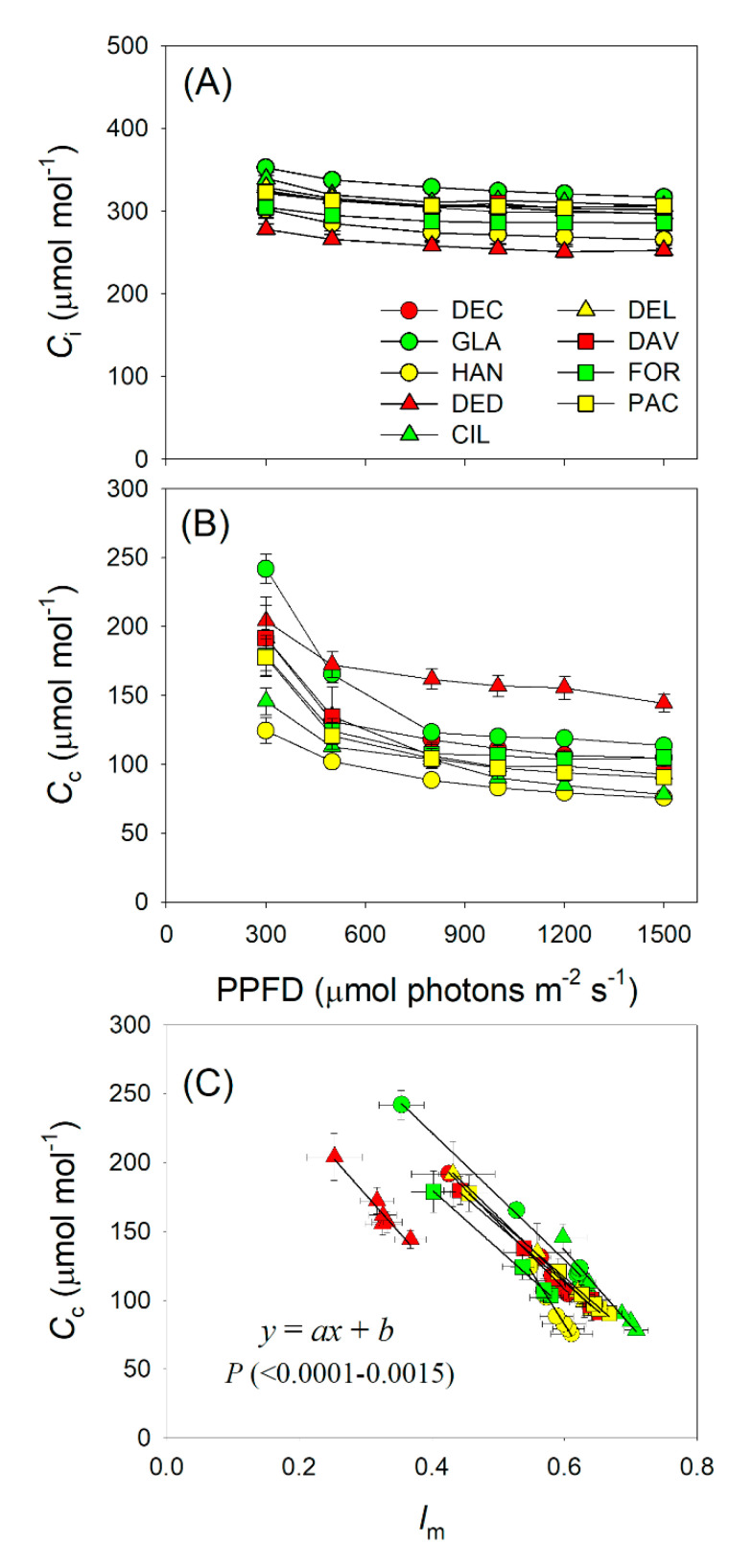
(**A**,**B**) Responses of intercellular and chloroplast CO_2_ concentrations (*C*_i_ and *C*_c_, respectively) to incident light intensity in nine *Rhododendron* species. (**C**) Relationship between *l*_m_ and *C*_c_ across light response curves in these nine *Rhododendron* species. Symbols represent means ± SE (*n* = 4–5). Species abbreviations: DEC, *R. decorum*; GLA, *R. glanduliferum*; HAN, *R. hancockii*; DED, *R. decorum* subsp. *disprepes*; CIL, *R. ciliicalyx*; DEL, *R. delavayi*; DAV, *R. davidii*; FOR, *R. fortunei*; PAC, *R. pachypodum*.

**Figure 6 plants-09-01536-f006:**
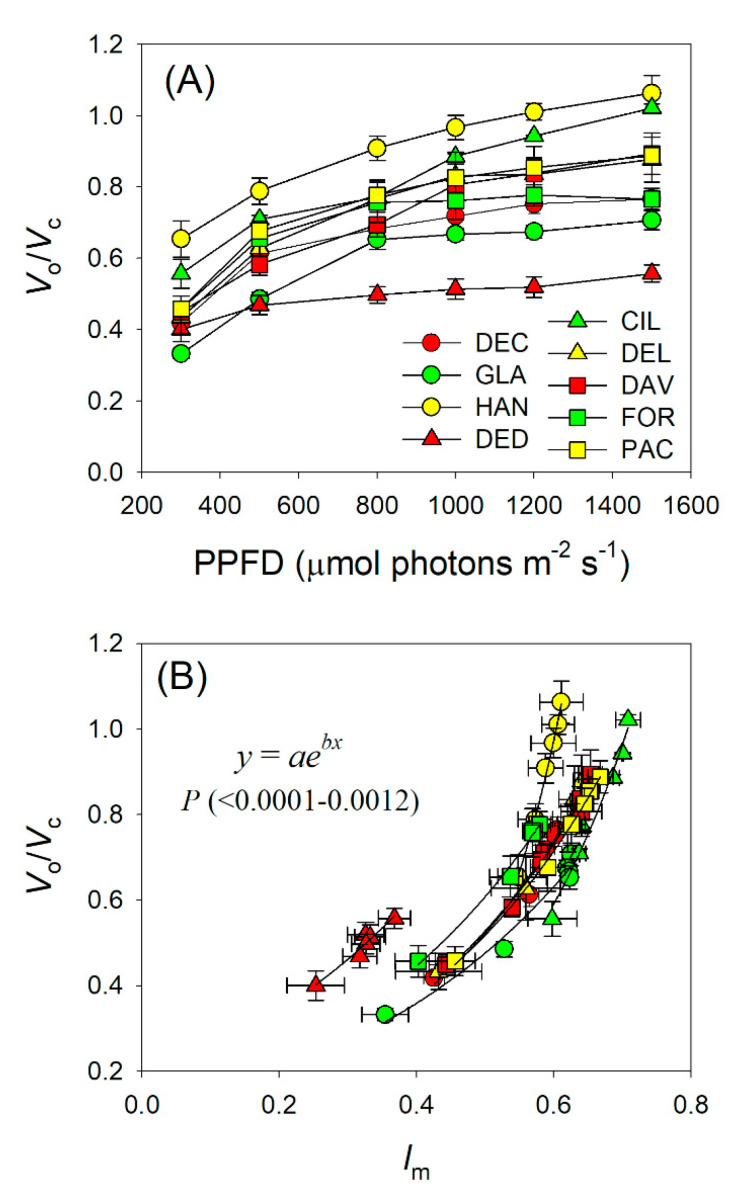
(**A**) Light intensity dependence of the *V*_o_/*V*_c_ ratio in nine *Rhododendron* species. (**B**) Relationship between *l*_m_ and *V*_o_/*V*_c_ across light response curves in these nine *Rhododendron* species. Symbols represent means ± SE (*n* = 4–5). Species abbreviations: DEC, *R. decorum*; GLA, *R. glanduliferum*; HAN, *R. hancockii*; DED, *R. decorum* subsp. *disprepes*; CIL, *R. ciliicalyx*; DEL, *R. delavayi*; DAV, *R. davidii*; FOR, *R. fortunei*; PAC, *R. pachypodum*.

**Figure 7 plants-09-01536-f007:**
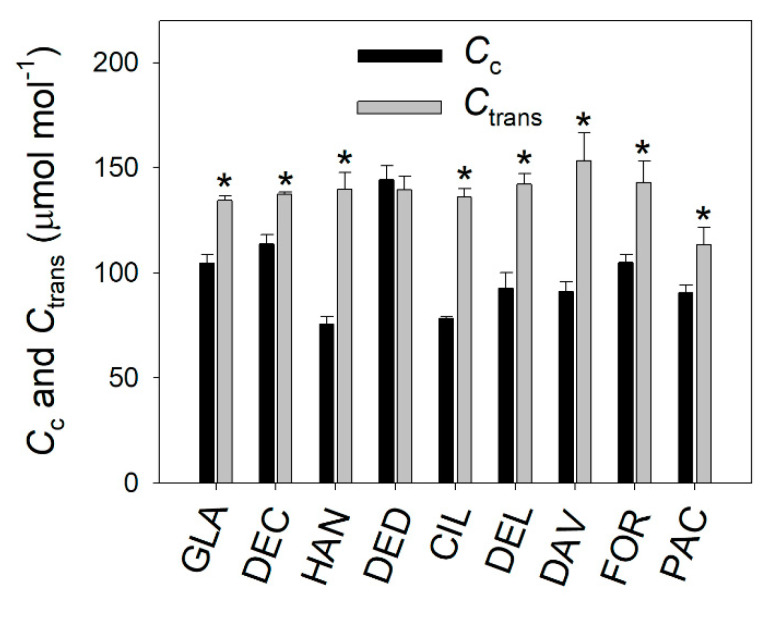
Comparison of *C*_c_ and *C*_trans_ at 1500 μmol photons m^−2^ s^−1^ in nine *Rhododendron* species. Symbols represent means ± SE (*n* = 4–5). Significant differences at a 5% confidence level are indicated with an asterisk. Species abbreviations: DEC, *R. decorum*; GLA, *R. glanduliferum*; HAN, *R. hancockii*; DED, *R. decorum* subsp. *disprepes*; CIL, *R. ciliicalyx*; DEL, *R. delavayi*; DAV, *R. davidii*; FOR, *R. fortunei*; PAC, *R. pachypodum*.

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
