# Peer review of "The Light Dependence of Mesophyll Conductance and Relative Limitations on Photosynthesis in Evergreen Sclerophyllous Rhododendron Species"

_plants, 2020, doi:10.3390/plants9111536_

Round 1
Reviewer 1 Report
See attached file.

Author Response
Responses to Reviewer 1
First, I’d like to thank the authors for a well-written manuscript. The English is quite good and that makes reviewing it much easier. Also, the manuscript is of the appropriate length for this study and the graphics are generally quite good, although I do have some comments on them (see below). My main criticisms are two: (1) I’d like to see the authors provide more specific hypotheses at the end of the Introduction, and (2) I think the authors should consider anatomical aspects of sclerophyllous leaves in more detail for why such plants generally have low gm compared to herbaceous/crop species.
I would consider revising the title. As is, the title refers only to gm, but much of the paper is devoted to relative limitations on An. Perhaps revise to: “The light dependence of mesophyll conductance and relative limitations on photosynthesis in …..” Just a suggestion so the title better reflects the content therein.
Response: We have changed the title into “The light dependence of mesophyll conductance and relative limitations on photosynthesis in evergreen sclerophyllous Rhododendron species”.
I would also like to know more about how R. decorum and R. decorum subsp. disprepes differ from each other. Do they co-occur in nature, or are they separated geographically? What exactly differs between them?
Response: R. decorum and R. decorum subsp. disprepes co-occur in nature. R. decorum subsp. disprepes has larger leaves and flowers.
Did the authors make any anatomical measurements of the leaves of each of their species, such as specific leaf area, thickness, percent internal airspace, cell wall thickness, stomatal densities, etc.?? This would help to put their results into context by allowing them to relate their findings to leaf anatomical characteristics. It might explain why R. decorum spp. disprepes seems so different from the other species.
Response: In this study, we did not make any anatomical measurements for leaves of these Rhododendron species. We are also wonder why R. decorum subsp. disprepes showed much higher gm than other species, and the cell-level anatomical characteristics will be conducted in the near future.
Along these same lines, can the authors provide more background information on the species examined? Their natural range, whether they grow beneath forest canopies in nature, near streams, etc. A better understanding of their ecology would help place their responses into better context. The authors state that all are “evergreen”, but does that mean they all have leaves that last more than one year, or does it mean that the shrubs have leaves all year round, but individual leaves do not live beyond one year? Both cases would be considered evergreen.
Response: These evergreen Rhododendron species have leaves that last more than one year.
In section 4.6 of the Methods, did the authors check whether the assumptions for parametric ANOVA were met, specifically, equality of variances?
Response: We have checked the homogeneity of variances, and found that the P values were higher than 0.05.
In the legend to Figure 2, it is not apparent what panel B is showing. Is this the average gm across all light values from each light response curve for a species as shown in Panel A? If so, it is partially redundant with Panel A, since nearly the same information can be obtained from that graph. However, I’m ok leaving it in if the editors agree.
Response: Indeed, Figure 2B showed the average gm across all light values from each light response curve for a species as shown in Panel A. We have deleted it in the revised version.
For Figure 4, please provide the equations used to fit the curves. There is a typo in the legend (take out the “t” before “nine”.
Response: We have added the equations in Figure 4.
Regarding my training in statistics, I was taught that if there is not a statistically significant relationship between two variables (e.g., panel B, triangles) you do not put a line through the, points, as that biases the reader into thinking there really is a relationship. Therefore, please remove the line through the triangles in Panel B, Figure 4.
Response: We have deleted the line through the triangles in Figure 4B.
Furthermore, the authors need to add explanation to Section 4.6 on how they constructed the regression relationships in Figures 4 and 5, since this is a special case of regression analysis where both the independent and dependent variables have “error” associated with them. I suggest consulting with a statistician to get this done correctly. I don’t think it will change their biological conclusions at all, but we should get the statistics correct. Include p values for regressions in Figure 5 (a range perhaps, as done for r2). See: https://doi.org/10.1590/S0100-40422013000600025
Response: We have revised the section statistical analysis and added p values in Figure 5.
Is there any reason why a regression wasn’t performed for data in Panel B, Figure 6? A non-linear relationship seems appropriate here.
Response: We have added non-linear regression in Figure 6B.
I am very skeptical of regressions driven by just one point, as shown in Figure 7. If the outlying points are eliminated in Panels B and C, Figure 7, I doubt there are any significant relationships there. I think this means that caution is called for when interpreting these results. What is different about the species characterized by that outlying point? It seems all the others group together with no relationship. Therefore, I am reluctant to say this represents a trend here. I think the authors should interpret their results without that outlying point, and then compare that with what they would conclude by including that point, and if they can, justify using that point if they can do so.
Response: We have deleted Panels B and C in Figure 7. The section Results and Discussion have been revised.
With regard to p values, I would restrict the number of decimal places to no more than three. Panel B in Figure 7 shows 4 decimal places, but Panel C only three. Please check on this throughout the paper and make all decimal places consistent at three only. No useful information is gained by having a fourth decimal.
Response: We have deleted panels B and C in Figure 7.
On line 176, section 3.1, the title of this section is “Rapid response…”. But this paper is not a study of how gm changes with time, but rather, with light. So the adjective “rapid” is not called for. Take out that word. Same problem with line 180. Gm limitation may respond rapidly to changes in light intensity, but that was not tested in this study. Best to say it responds to changes in light intensity and to do another study of the time course of change if that is of interest. The word “rapid” is used repeatedly throughout this section and in the paper, and I suggest deleting it everywhere it is used. If light causes changes in RuBP activation state, that takes time, as RuBISCO activase is involved, among other things, and therefore any changes in gm will occur over some finite time period, which may differ depending on the preceding length of the low light period, and vice-versa. All of this means that assessing how “rapid” gm changes is inappropriate for this paper. Best to just say it differs depending on light intensity, without reference at all to time.
Response: Thanks a lot for this important comment. We have delete “rapid” throughout the text.
Overall, the paper reports interesting findings regarding the light dependency of gm limitation on An. It would add to the paper if the authors could provide additional information about R. decorum spp. diaprepes, as it seemed quite different from all the others. Why?
Response: This important comment raises a question that why R. decorum subsp. diaprepes has much higher gm than other species. In order to answer this question, we will examine the anatomical characteristics related to gm, including Sc and Sm.
Lastly, the authors compare their results in the Discussion section largely to herbaceous species, mainly crop species, and it might help if they could include references to other sclerophyllous species, as the question of why sclerophyllous species in general have lower saturating An than other species has been the subject of investigation for quite a while. There are several recent papers out just this year and last that look into other possibilities for low gm in such species that include such ideas as tortuous internal diffusion of CO2 so that it is more difficult to get CO2 to the chloroplasts. This often involves differences in cell density and cell wall thickness. Adding in a section discussing these attributes would improve the Discussion and show that some limitations may be anatomical in origin. The evolutionary tradeoff of making an evergreen, sclerophyllous leaf may be that the structural investment to withstand adverse winter conditions precludes evolving high gm.
Response: We have revised the section Discussion according to these important comments.
Grammatical suggestions
Line 44 – take out “some” at end of line; not needed
Line 50 – change “are” to “is”, singular form, as it refers to “light response”, which is singular
Line 59 – somewhat awkward as written; I suggest changing to “In addition to RuBP carboxylation, ….”
Line 60 – sentence ends without clarifying “environment”. Rewrite as: “…under current atmospheric environmental conditions.”
Lines 65 and 226 – I disagree with the wording here. Photorespiration is probably happening even at low light, just at low rates. The word “triggering” makes it sound like photorespiration is suddenly turned on. I’d replace “triggering” with “enhancing”.
Line 66 – see also older references to H. Max Vines, who did pioneering work on the relationship between light intensity and photorespiration in plants. Consider citing these, since he did this before anyone else back in the early to mid-1970s.
Line 72 – It is not clear what is meant by “conversely”. If the authors mean when Cc is higher than Ctrans, then they should rewrite as: “In contrast, when the opposite is true, i.e., Cc is higher than Ctrans, then An tends to ….”. On line 233, the same problem occurs, but the authors use the phrase “on the contrary)”.
Line 78 – I would like the authors to come up with hypotheses that are more specific. After explaining various scenarios that can occur in plants, I think the authors could develop better hypotheses to test rather than just saying that gm and lm affect An.
Line 170 – Reference to Panel C is missing from legend to Figure 7.
Line 171 – change “were” to “are”
Line 198 – change “is” to “are”, as you referring to two different things (growth and productivity).
Line 219 and 220 – did the authors mean to write “photorespiration”? Did they mean “photosynthesis”?
Line 235 – change “were” to “was” since “light” is singular, or, to be consistent with the end of the sentence, which is in the plural, rewrite as: “…values of operating Cc..were….”
Line 236 – is “diaprepes” misspelled? Should it be “disprepes”?
Line 247 – rewrite as: “…resulting in Cc being lower than Ctrans.”
Line 262 – I would change as follows: “…using the 2-cm2 measuring head…”
Line 285 – rewrite as: “…absence of daytime respiration [41,48 and we used a typical value of 40 umol mol-1 in our current study [19.”
Line 287 – take out “the” at the beginning of the line. Not necessary.
Line 296 – Can 406.8 umol mol-1 be written as 407, since 0.2 umol/mol is probably too high a precision for such calculations?
Line 297 – Units are missing for concentration of “O”.
Line 298 – It is not clear how the rate-limiting step was determined just from reading this sentence. If it refers to the next section, perhaps tell the reader that.
Line 316 – change “test” to “tests”, since this should be plural.
Response: Thanks a lot for these suggestions. These grammatical suggestions were carefully revised through the text.

Reviewer 2 Report
The study by Yang et al. entitled “The light response of mesophyll conductance and its effect on photosynthesis in evergreen sclerophyllous Rhododendron species” presents mesophyll conductance data related to light in different species of the same genus, all possessing sclerophyllous and evergreen leaves. I enjoyed the length of the paper, which concisely presented their dataset and made an appropriate discussion of them.
I would like to point out that I feel the authors understated the novelty of their study. They present the other studies that have presented gm-light response, but they do not sufficiently highlight that their study deals with sclerophyllous and evergreen leaves. This is the novelty, and I would highly suggest rewriting parts of the introduction with that focus in mind, and this would also affect how the discussion and conclusion.
I have also a few concerns regarding the analysis. First of all, the light response was measured down to 300 PPM of light, a value at which the majority of species were light-saturated or almost saturated (see Fig. 1). For me, concluding that there is no response to light is obvious because the leaves were light-saturated, hence expecting no response at all. The manuscript does not mention that fact as far as I know. This should be acknowledged, and by doing so the discussion would substantially change in my opinion.
Secondly, for a few analyses, one species pulls the regression. I have concerns that these regressions would become nonsignificant without that point. As a consequence, I suggest the authors redo the analyses on those species only (the ones with positive Ctrans-Cc). This would bring a more convincing argument in my opinion.
You will find below other comments, some of which might be repeated above.
I hope the authors will find my comments useful and that they will allow improving the manuscript.
Sincerely,
A reviewer
-----
L28-30: This is too general in my opinion. I think the specificity of this study is the focus on several sclerophyllous species of the same genus. I would thus think starting with differentiating sclerophyllous and/or evergreens would be more appropriate to set the tone of the paper. Gm could be introduced in that sentence or in the second.
L38: I wouldn't say hours are rapid, so I would leave out rapid: it's clear that seconds and minutes are rapid.
L48-49: This sentence is out of place. I would move it to the last paragraph since you do not even mention Rhododendron before then. Further, the light response of gm is poorly understood in plants in general, so I would specify this. Rather, you can mention that it has been understudied in evergreen and/or sclerophyllous leaves, which differ in construction and properties compared to herbaceous plants and deciduous tree leaves.
Results, section 2.1: Mentioning the saturation point of each species would be relevant. Rhododendron are often sub-canopy species thus grow at relatively low light, and Figure 1 suggests that they saturate around 300 PPM light.
Figure 2: Rather than creating two panels, the averages could be added as points to the right of panel A (e.g. where 1800 PPFD is) or as values next to the labels in the legend. Just an idea. My point is that the bar plot doesn't show much more than the light responses.
L153-154: These are very weak relationships as only one point is dragging the correlation. I would suggest running the correlation on all species that had positive Ctrans-Cc to check if the relationship holds and how significant it is.
Discussion, section 3.1: You refer to [17]. This study highlighted that the gm-light response is an apparent response caused by a heterogenous light profile within the leaf. If you didn't observe gm light response this would suggest that your leaves were light-saturated, which seems the case based on Fig.1. This study was also one of the few that measured below 300 PPFD. So it cannot be ruled out that the lack of response observed here is because leaves were saturated in light. Had you went to lower light intensities, i.e. below 300 in the non-saturating zone, I would have expected a decrease in gm (see lines L188-189).
L192-193: This should be expanded on if it is left in the manuscript. I'm not sure I follow what you have in mind. What is meant to be conserved here? Water?
L255: Are the plants growing under full sunlight? The usual light regime should be presented in my opinion.
L321-324: As I mentioned earlier, 300 was saturating or almost saturating light for all species. So it is normal that no gm response to light is observed. So I wouldn't say this is an accurate conclusion. More needs to be detailed to back up that statement and counter-arguments presented as pointed out in the studies you have cited.
As for the conservative strategy: is it the response of gm that is conservative, or the light response of photosynthesis that rapidly plateaus?
Author Response
Responses to Reviewer 2
The study by Yang et al. entitled “The light response of mesophyll conductance and its effect on photosynthesis in evergreen sclerophyllous Rhododendron species” presents mesophyll conductance data related to light in different species of the same genus, all possessing sclerophyllous and evergreen leaves. I enjoyed the length of the paper, which concisely presented their dataset and made an appropriate discussion of them.
I would like to point out that I feel the authors understated the novelty of their study. They present the other studies that have presented gm-light response, but they do not sufficiently highlight that their study deals with sclerophyllous and evergreen leaves. This is the novelty, and I would highly suggest rewriting parts of the introduction with that focus in mind, and this would also affect how the discussion and conclusion.
Response: Thanks a lot for this important comments. We have added the novelty of this study in the section Introduction (please see lines 60-62). Previous studies mainly focused on the light response of gm in herbaceous plants. However, the response of gm to light intensity in sclerophyllous evergreen species is not well known.
I have also a few concerns regarding the analysis. First of all, the light response was measured down to 300 PPM of light, a value at which the majority of species were light-saturated or almost saturated (see Fig. 1). For me, concluding that there is no response to light is obvious because the leaves were light-saturated, hence expecting no response at all. The manuscript does not mention that fact as far as I know. This should be acknowledged, and by doing so the discussion would substantially change in my opinion.
Response: Thank the reviewer for this suggestion. We have checked whether AN is saturated at 300 μmol photons m−2 s−1 by calculating the A300/A1500 ratio. In these species, the A300/A1500 ratios ranged from 0.64 to 0.79, indicating that AN was not saturated at 300 μmol photons m−2 s−1.
Secondly, for a few analyses, one species pulls the regression. I have concerns that these regressions would become nonsignificant without that point. As a consequence, I suggest the authors redo the analyses on those species only (the ones with positive Ctrans-Cc). This would bring a more convincing argument in my opinion.
Response: We have deleted panels B and C in Figure 7.
L28-30: This is too general in my opinion. I think the specificity of this study is the focus on several sclerophyllous species of the same genus. I would thus think starting with differentiating sclerophyllous and/or evergreens would be more appropriate to set the tone of the paper. Gm could be introduced in that sentence or in the second.
Response: We have revised these sentences (please seen lines 32-34). In Southwest China, mountain forest ecosystems are characterized by the presence of evergreen sclerophyllous angiosperms. This vegetation type includes many species of the genus Rhododendron (Ericaceae) that exhibit low mesophyll conductance (gm).
L38: I wouldn't say hours are rapid, so I would leave out rapid: it's clear that seconds and minutes are rapid.
Response: We have deleted “hours”.
L48-49: This sentence is out of place. I would move it to the last paragraph since you do not even mention Rhododendron before then. Further, the light response of gm is poorly understood in plants in general, so I would specify this. Rather, you can mention that it has been understudied in evergreen and/or sclerophyllous leaves, which differ in construction and properties compared to herbaceous plants and deciduous tree leaves.
Response: We have moved this sentence to the first paragraph (see lines 32-34). In Southwest China, mountain forest ecosystems are characterized by the presence of evergreen sclerophyllous angiosperms. This vegetation type includes many species of the genus Rhododendron (Ericaceae) that exhibit low mesophyll conductance (gm).
Results, section 2.1: Mentioning the saturation point of each species would be relevant. Rhododendron are often sub-canopy species thus grow at relatively low light, and Figure 1 suggests that they saturate around 300 PPM light.
Response: In these species, the A300/A1500 ratios ranged from 0.64 to 0.79, indicating that AN was not saturated at 300 μmol photons m−2 s−1. The A800/A1500 ratios ranged from 0.86 to 1.02, suggesting that AN were saturated or almost saturated at 800 μmol photons m−2 s−1. We have added this sentence in Results section 2.1 (please seen lines 109-110).
Figure 2: Rather than creating two panels, the averages could be added as points to the right of panel A (e.g. where 1800 PPFD is) or as values next to the labels in the legend. Just an idea. My point is that the bar plot doesn't show much more than the light responses.
Response: We have deleted panel B in the revised version.
L153-154: These are very weak relationships as only one point is dragging the correlation. I would suggest running the correlation on all species that had positive Ctrans-Cc to check if the relationship holds and how significant it is.
Response: The panels B and C in Figure 7 have been deleted in the revised version.
Discussion, section 3.1: You refer to [17]. This study highlighted that the gm-light response is an apparent response caused by a heterogenous light profile within the leaf. If you didn't observe gm light response this would suggest that your leaves were light-saturated, which seems the case based on Fig.1. This study was also one of the few that measured below 300 PPFD. So it cannot be ruled out that the lack of response observed here is because leaves were saturated in light. Had you went to lower light intensities, i.e. below 300 in the non-saturating zone, I would have expected a decrease in gm (see lines L188-189).
Response: First, in these Rhododendron species AN were not saturated at 300 μmol photons m−2 s−1. Second, even AN was saturated, gm still gradually increased with increasing light intensity in Populus nigra and Oryza sativa (Xiong et al. 2018). Therefore, the response of gm to light intensity is species dependent.
L192-193: This should be expanded on if it is left in the manuscript. I'm not sure I follow what you have in mind. What is meant to be conserved here? Water?
Response: We have revised this sentence (see lines 196-197). Therefore, in order to optimize the cultivation strategy, we should take into consideration the relative limitations of photosynthesis under different light intensities.
L255: Are the plants growing under full sunlight? The usual light regime should be presented in my opinion.
Response: Yes, these plants are exposed to full sunlight.
L321-324: As I mentioned earlier, 300 was saturating or almost saturating light for all species. So it is normal that no gm response to light is observed. So I wouldn't say this is an accurate conclusion. More needs to be detailed to back up that statement and counter-arguments presented as pointed out in the studies you have cited.
Response: As mentioned above, AN were not saturated at 300 μmol photons m−2 s−1.
As for the conservative strategy: is it the response of gm that is conservative, or the light response of photosynthesis that rapidly plateaus?
Response: This is an attractive question. In my opinion, the response of gm to light may be a conservative trait for species in a given genus. The response of gm to light intensity may affect the light saturating point. For example, if gm gradually increases with light intensity, AN may be saturated at a higher light intensity.

Round 2
Reviewer 2 Report
I generally enjoyed the new version and was pleased by the edits. Removing certain paerts of figures was a good choice to remove potential confusion. I was able to detect a few points I think are important to clarify, especially regarding the wording of one of the main conclusions. Also, I think highlighting what Rhododendron bring new should be briefly mentioned as previous work has been done on several evergreen species, about 50-50. I am sure the author have a clear idea, for example that they are understory trees, but this is not obvious in the text.
Below are my comments, which might repeat some of the comments above. As there were no line numbers, I tried to be a precise as possible with their location.
I hope these comments will help finalize this concise and well written manuscript.
Sincerely,
A reviewer
----
- 2, end of 2nd paragraph, as well as p.5, section 3.1, last sentence of 1st paragraph: The previous studies you cite are about 50-50 herbaceous and woody species, including evergreens (Camellia, Arbutus, and Eucalyptus). The two last sentences of that paragraph need to be revised as they do not reflect the current state of the literature. What does Rhododendron has that the previously studied species do not?
p.3, 2nd paragraph: there are two W. Yamori et al. 2010 references in the same line.
p.3, section 2.1 (also Methods section 4.3): Please specify which method of Harley. I assume the variable J method was used, but it needs to be specified.
p.3, last sentence of page: I think removing “light response” in tat sentence would remove confusion. Since it is said that thre were no significant changes before, writing this can suggest to the less attentive reader that there is average light response, not an average value. Also, it is mentioned that there are no significant changes, but no statistics are shown. This needs to be adressed.
p.4, first paragraph of section 2.3: The variable J method, which I assume was used to estimate gm, is sensitive to variations in Ci. So with the range of values observed between species, there might have been a Ci effect on gm estimates, especially when comparing gm values at 300 and 1500 umol light. If the authors could compare gm estimate within a 10 ppm range in Ci that might help in the analysis, though I assume that Ci values for points measured above 800 µmol light should be in a narrow range and thus not exhibit a Ci effect.. I would suggest the authors to look into that and adjust the text if appropriate.
p.5 2nd paragraph of section 3.1: I would rephrase one sentence. You should highlight that there is a lack of response rather than talking about rapid response. Here in “we propose that the rapid response of gm to the change in light intensity…” I woulkd instead write “we propose that the lack of response of gm to…” or something similar. Samewise, in p.9, Conclusions, I would say “Therefore, the lack of response of gm to rapid changes…”, not just “the response”, because this will highlight what is exactly the conservative strategy, i.e. no variation under environmental changes. This actually makes a lot of sense for understory vegetation and shade leaves in general in order to maintain carbon assimilation under varying light regimes.
Figure 2: I would have the y-axis go up to 0.25 only to have a bit more spread in the values tucked between 0.05 and 0.1. Also, the average value of DED seems wrong at 0.08.
Figure 5A: I would highly recommend to have the y-axis go only from 200 to 400 to have a better look at the difference between species and have a better idea of the raw values.
References: The Yamori references should be double checked.